# Exercise routine change is associated with prenatal depression scores during the COVID-19 pandemic among pregnant women across the United States

**Theresa E. Gildner** [1,2]*, **Elise J. Laugier** [1,3], **Zaneta M. Thayer** [1,3]

1 Department of Anthropology, Dartmouth College, Hanover, New Hampshire, United States of America,
2 Department of Anthropology, Washington University, St. Louis, Missouri, United States of America,
3 Ecology, Evolution, Environment & Society Program, Dartmouth College, Hanover, New Hampshire, United States of America

* Theresa.E.Gildner@dartmouth.edu

**Data Availability Statement:** CARE Study Data Sharing Policy: The CARE study dataset contains sensitive and potentially identifying participant information. Ethical and legal restrictions therefore

## Abstract

### Background

The COVID-19 pandemic has negatively affected physical and mental health worldwide. Pregnant women already exhibit an elevated risk for depression compared to the general public, a pattern expected to be exacerbated by the pandemic. Certain lifestyle factors, including moderate exercise, may help support mental health during pregnancy, but it is unclear how the pandemic may impact these associations across different locations. Here, we test whether: (i) reported exercise routine alterations during the pandemic are associated with depression scores; and, (ii) the likelihood of reporting pandemic-related exercise changes varies between women living in metro areas and those in non-metro areas.

### Methods

This cross-sectional study used data from the COVID-19 And Reproductive Effects (CARE) study, an online survey of pregnant women in the United States. Participants were recruited April-June 2020 (n = 1,856). Linear regression analyses assessed whether reported COVID-19-related exercise change was associated with depression score as measured by the Edinburgh Postnatal Depression Survey. Logistic regression analyses tested whether a participant's Rural-Urban Continuum Code classification of "metro" was linked with higher odds of reporting exercise changes compared to a "non-metro" classification.

### Results

Women who reported exercise changes during the pandemic exhibited significantly higher depression scores compared to those reporting no changes. Moreover, individuals living in metro areas of all sizes were significantly more likely to report exercise changes compared to women living in non-metro areas.

prevent us from posting our data to a third-party server. However, data and statistical code will be made available to qualified researchers upon reasonable request. Researchers must agree to privacy and data use expectations that conform to IRB requirements and are central to the welfare of study participants. Requests for data can be easily made using a request form on the study website (https://sites.dartmouth.edu/care2020/request-data-access/).

**Funding:** Dr. Zaneta Thayer was funded by the Wenner-Gren Hunt Fellowship (grant #9687) during this project. Participant compensation was provided through the Claire Garber Goodman Fund in the Department of Anthropology at Dartmouth College.

**Competing interests:** The authors have declared that no competing interests exist.

## Conclusions

These results suggest that the ability to maintain an exercise routine during the pandemic may help support maternal mental health. It may therefore be prudent for providers to explicitly ask patients how the pandemic has impacted their exercise routines and consider altered exercise routines a potential risk factor for depression. An effort should also be made to recommend exercises that are tailored to individual space restrictions and physical health.

## Introduction

The COVID-19 pandemic has impacted the health and livelihoods of people around the world [1]. However, certain countries have been hit harder than others, with the United States exhibiting the highest prevalence and mortality estimates globally. As of November 16, 2020, 54,563,236 people living in the U.S. are estimated to have been infected with the novel coronavirus SARS-CoV-2, while 246,526 are thought to have died from the resulting disease COVID-19 [1]. Within the U.S., certain populations appear to be at higher risk for negative COVID-19 disease outcomes, including pregnant women. Although the specific effects of COVID-19 on pregnancy are still being tested, some researchers argue pregnant women are generally at higher risk for viral respiratory illness [2]. Moreover, other studies indicate that COVID-19 may be associated with an elevated risk for intensive care unit admission, placental injury, pre-eclampsia, preterm birth, low birth weight, and even maternal death [3–6]. These recent findings led the CDC to add pregnancy as a risk factor for severe COVID-19 outcomes on June 25, 2020 [7].

In addition to potentially being at an elevated risk for poor COVID-19 health outcomes, the CDC cautions that pregnant women may feel increased stress or anxiety during the pandemic [7]. The physical distancing and "stay at home" orders implemented by many local governments to slow the spread of SARS-CoV-2 are thought to increase feelings of anxiety and isolation, especially among already vulnerable populations [8]. Pregnancy-associated physiological changes have been clearly linked with increased depression risk, such that 1 in 8 women in the U.S. have been estimated to experience postpartum depression symptoms [9]. These depression-related symptoms include fatigue, changes to appetite or sleep, crying more often than usual, withdrawing from loved ones or the baby, feelings of anger, sorrow, hopelessness, worthlessness, or restlessness, and suicidal ideology [9]. Recent evidence suggests that maternal depression symptomatology has become even more common during the COVID-19 pandemic [8, 10].

Specifically, one study comparing mental health among pregnant women/new mothers before and after the onset of the pandemic found that rates of clinical depression and moderate to high anxiety significantly increased with the onset of the pandemic [8]. These changes appear to be directly linked with the disruptions to daily routines, social isolation, and fears of developing COVID-19, highlighting the need for increased maternal health screening and treatment [8]. Other factors, including increased COVID-19-related financial stress or changes to working plans in pregnancy have also been associated with increased depression [11, 12]. Given the probable exposure of offspring to any psychotropic medications taken by the mother during pregnancy or breastfeeding [13], the identification of non-pharmaceutical treatments to reduce maternal depression is of great interest. One such alternative treatment may be maintaining a regular exercise routine throughout pregnancy and the postpartum period, which has been shown to both prevent and help treat depression symptoms among women able to safely engage in moderate to vigorous physical activity [13–20].

Regular exercise has been linked with several beneficial effects, including improved body satisfaction [21], reduced physical discomfort [22], feelings of physical control despite somatic changes linked with pregnancy [23, 24], and improved mood through the release of endorphins and neurotransmitters [23, 25]. It has therefore been recommended by the American College of Obstetricians and Gynecologists that pregnant women without medical contraindications engage in at least 150 minutes of moderate-intensity aerobic activity every week, divided into 30 minutes intervals most days of the week [26]. Yet, many pregnant women do not exercise at the recommended levels, due in large part to fatigue, time constraints (e.g., work and childcare), and pregnancy-related physical limitations (e.g., back pain, swelling, and joint pain) [27, 28]. Differences are also apparent by location, with pregnant women living in urban areas exhibiting significantly higher levels of moderate physical activity compared to those living in rural areas, likely due in part to increased access to fitness centers [29, 30].

Regardless of location, the benefits of exercise during pregnancy appears to be especially relevant during the COVID-19 pandemic, with women who reported at least 150 min of moderate intensity exercise exhibiting significantly lower scores for both anxiety and depression compared to those who exercised less [8]. Yet the direct effects of the pandemic on exercise regimens among pregnant women has not been fully explored. For instance, within the United States, shelter in place orders were implemented to some extent across nearly all states throughout the spring of 2020 [31]. It therefore seems likely that many women's exercise routines have been disrupted by the pandemic, especially in metro areas (e.g., due to fitness centers and parks closing or a fear of viral exposure while exercising outside in a densely populated area). Conversely, women in less densely populated areas may feel safer going outside to engage in physical activities. However, the association between geographic location and the likelihood of reporting pandemic-related exercise changes has yet to be directly assessed. To the best of our knowledge, no other study has directly tested these relationships during the COVID-19 pandemic. In other words, it is not clear if reported pandemic-related changes in exercise routines are significantly associated with depression risk, and whether geographic location may influence the likelihood of these exercise disruptions.

We therefore use data drawn from the COVID-19 and Reproductive Effects (CARE) study–an online survey of pregnant women living in the U.S. which assesses how the COVID-19 pandemic has affected pregnant women's wellbeing–to test the following two hypotheses:

1. Whether reported exercise routine change during the COVID-19 pandemic was significantly associated with participant depression score.

2. Whether the likelihood of reporting pandemic-related changes in exercise routine significantly differed between women living in a metro area and those in a non-metro area.

## Materials and methods

### Study design

The COVID-19 And Reproductive Effects (CARE) study was posted on social media platforms (Facebook, Twitter) and distributed via email to contacts working in maternity care and public health. Pregnant women over the age of 18 and living in the U.S. were invited to participate in a short survey assessing how the COVID-19 pandemic had impacted their medical care and birth plans. The data presented here were collected between April 16 –June 16, 2020. This study received ethical approval from Dartmouth College (STUDY00032045).

The survey was administered using REDCap (Research Electronic Data Capture) hosted through Dartmouth College. REDCap is a secure web platform which facilitates the creation

and management of online surveys for research studies [32, 33]. Only women who met the inclusion criteria (i.e., living in the U.S., over 18 years of age, and currently pregnant) were eligible to participate in the survey. The survey completion rate (i.e., the percentage of those who consented to take the survey and actually went through to the end of the questionnaire) was 80%. During the study period, there were 1,856 surveys collected that included responses for all study variables. Data on depression symptomatology, exercise routines, and geographic location were collected, along with other covariates known to influence exercise patterns and depression risk.

**Depression scores.** Depression symptoms were screened using the Edinburgh Postnatal Depression Survey (EPDS), resulting in a participant score ranging from 0 (minimum) to 30 (maximum) [34]. Depression scores were analyzed continuously.

**Exercise routine changes.** Participants were asked "has your exercise routine changed at all since the COVID-19 pandemic began?" Participants responded yes or no (making this a dichotomous variable). Additionally, participants were asked the number of days per week (on average) they engaged in moderate exercise for at least 30 minutes.

**Geographic location.** Participants self-reported their zip codes. This information was used to generate a Rural-Urban Continuum Code (RUCC) for each respondent [35]. The RUCC is based on the county located at the zip code center point, with each participant receiving a code on a 1–9 scale. For this analysis, the codes were collapsed as follows: 1) metro county of 1 million people or more, 2) metro county of 250,000 to 1 million people, 3) metro county of less than 250,000 people, and 4) all non-metro counties. In addition, participants reported whether or not they lived in an area where they were currently required to shelter in place (i.e., not leave home, except for essential activities).

**Age.** Past research indicates that maternal age is inversely related to depressive symptoms and exercise frequency during pregnancy [36, 37]. Thus, participants self-reported their age in years and this variable was included in the statistical models.

**Current gestational week.** Gestational week has been shown to influence depression risk; for instance, women in one study exhibited fewer depressive symptoms during the middle of their pregnancy [38]. In addition, physical changes (e.g., swollen joints and back pain) that may impair ability to exercise normally are generally more common later in pregnancy [39]. Thus, participants indicated their current gestational week in the survey, and this variable was included during analysis.

**Race/Ethnicity.** Race/ethnicity has been linked with maternal depression risk, with minority populations exhibiting higher depression rates [36]. Moreover, racial/ethnic differences in beliefs about the safety of exercise during pregnancy have been documented, such that feeling unsafe/unsure about exercise during pregnancy was associated with non-white race/ethnicity, which may subsequently influence exercise patterns during pregnancy [40]. Thus, participant race/ethnicity was self-reported and measured according to the Office of Management and Budget Standards [41]. Native Hawaiian/Pacific Islander participants were re-classified as "Other" due to a small sample size (n = 3).

**Household income.** Previous work indicates that higher income levels may protect against maternal depression, while also enhancing access to expensive physical activity resources (e.g., personal exercise equipment and gym memberships) [36, 42]. Household income was consequently included in the statistical models. Participants indicated their household income from the following options: Less than $10,000 (1); $10,000 –$19,999 (2); $20,000 –$34,999 (3); $35,000 –$49,999 (4); $50,000 –$74,999 (5); $75,000 –$99,999 (6); $100,000+ (7). A composite household income variable was created for analysis: < $49,999, $50,000 –$99,999, and $100,000+.

**Education.**    Lower education levels have been linked with less exercise and increased depression risk during pregnancy [36, 43]. Education level was therefore included in the statistical models. Participants selected their highest completed education from the following options: Some high school, no diploma (1); High school graduate, diploma or the equivalent (for example: GED) (2); Some college credit, no degree (3); Trade/technical/vocational training (4); Associate degree (5); Bachelor's degree (6); Master's degree (7); Professional degree (8); Doctorate degree (9). A composite education variable was created for analysis: less than a bachelor's degree, a bachelor's degree, or a degree beyond a bachelor's degree.

**Financial stress.**    Financial stress appears to increase the likelihood of depression (including during the COVID-19 pandemic) and impedes access to some exercise options (e.g., decreases ability to afford a gym membership) [12, 42]. Participants were therefore asked to rate whether they agreed or disagreed (Likert question; 5 options ranging from strongly disagree to strongly agree) with the statement "I am worried about my financial situation due to the COVID-19 crisis." Participants who stated that they agreed or strongly agreed with the statement were classified as having experienced COVID-19-related financial stress.

**Risk status.**    Given that a high-risk pregnancy may increase depression risk and preclude engagement in certain forms of physical activity [44, 45], this variable was included during analysis. Women were asked whether their pregnancy was classified as "high-risk." In addition, participants aged 35 or older were classified as "high-risk".

## Statistical analysis

Data analyses were conducted using Stata 14. All continuous variables exhibited normal distributions, with skewness values within approximately ±0.5 and kurtosis values within approximately ±3. Multicollinearity was not detected between any variables; all VIF values were in an acceptable range of 1.01–1.65. Study descriptive statistics were calculated, and regression analyses were conducted to test the study hypotheses. Results were considered statistically significant at a p-value of less than 0.05.

**Hypothesis 1.**    Linear regression analyses assessed whether participants who report changes in their exercise routines during the COVID-19 pandemic demonstrated significantly higher depression scores compared to women who were able to maintain their exercise routine during the pandemic.

**Hypothesis 2.**    Logistic regression analyses tested whether participants living in RUCC classified metro areas (of all sizes) were significantly more likely to report COVID-19-related changes in their exercise routines compared to individuals who lived in non-metro areas.

Consistent with similar studies [12, 46, 47], regression analyses were used to test the hypotheses while controlling for covariates that may also be associated with the outcome of interest [48]. Given that depression score was a continuous dependent variable, linear regression was used to test the first hypotheses, while logistic regression was used to test the second hypothesis because reported exercise change was a dichotomous dependent variable [48]. All regression models adjusted for maternal age, education, income, financial stress, week of pregnancy at time of survey, race/ethnicity, and "high-risk" pregnancy.

## Results

### Sample characteristics and descriptive statistics

The study sample was spread out across all 50 states, including one participant in the U.S. territory of Puerto Rico (Fig 1). A clear majority of the sample (92.2%) indicated they lived in a location that required individuals to shelter in place (i.e., stay home, except for essential activities) at the time they completed the survey. Most women in the sample were physically active,

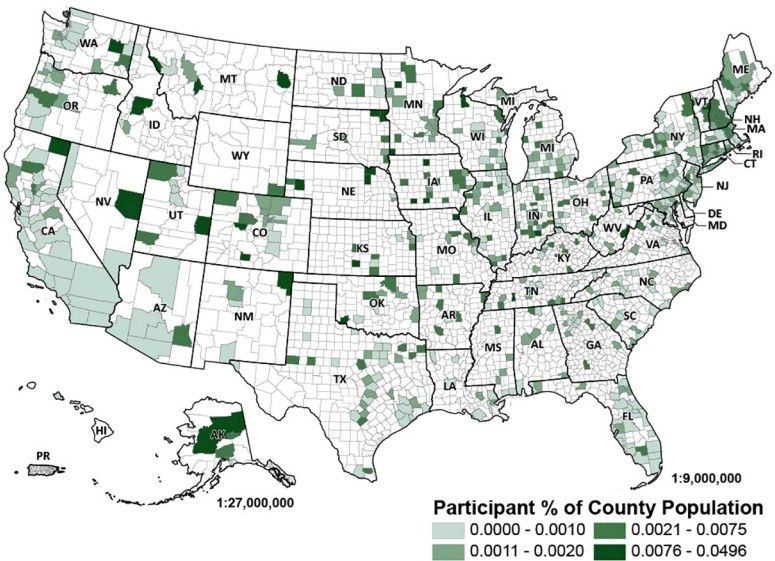

**Fig 1. Study participant locations across the United States.** The number of participants in each zip code is summarized across United States counties and displayed as a percentage of county population [35], with darker colors representing areas where participants make up a higher percentage of the county population (see legend). Reprinted from [35, 49] under a CC BY license, with permission from the U.S. Census Bureau, original copyright 2017, and the U.S. Department of Agriculture, original copyright 2013.

with 56.4% of respondents reporting they engaged in moderate exercise for at least 30 minutes three or more days per week (versus the 17.8% of women who reported they were inactive).

Sample descriptive statistics are presented in Table 1. Mean participant age was 31 years old and the mean number of weeks pregnant was 26. Most respondents were white (87% of the sample), educated (77% had at least a bachelor's degree), lived in a metro area of 1 million or more people (61% of the sample), and were relatively high-income (55% reported an annual household income of $100,000 or more). Still, approximately 43% of participants reported that they worried about their financial situation due to the COVID-19 pandemic. Most participants (65%) did not indicate a high-risk pregnancy. Additionally, the majority of participants (60%) reported that their exercise routine had changed during the COVID-19 pandemic. Finally, participants varied in depression scores as measured by the EPDS scale, ranging from the minimum score of 0 all the way to the maximum score of 30; the mean EPDS score was 10.6.

### Hypothesis 1

Linear regression analyses were carried out to assess whether exercise routine change during the pandemic was significantly associated with depression score, measured using the EPDS (Table 2). Participants who were older (B = -0.078, 95%CI: -0.148-(-0.009), p = 0.028), reported a higher household income (compared to < $49,999) of $50,000 - $99,999 (B = -0.861, 95%CI: -1.69-(-0.029), p = 0.042) or $100,000+ (B = -1.26, 95%CI: -2.14-(-0.390), p = 0.005), and were more highly educated (compared to less than a bachelor's degree) with either a bachelor's degree (B = -0.841, 95%CI: -1.52-(-0.157), p = 0.016) or a degree beyond a bachelor's degree (B = -1.08, 95%CI: -1.79-(-0.379), p = 0.003) exhibited significantly lower depression scores. Conversely, participants who were Hispanic, Latino, or Spanish in origin (B = 1.32, 95%CI: 0.332–2.30, p = 0.009), who reported a high-risk pregnancy (B = 0.573, 95% CI: 0.003–1.14, p = 0.049), or who reported experiencing COVID-19-related financial stress (B = 2.29, 95%CI: 1.81–2.78, p < 0.001) displayed significantly higher depression scores.

**Table 1. Descriptive statistics of model variables.**

| Variable | Mean (SD; range) |
|---|---|
| Age (years) | 31.3 (4.30; 18–47) |
| Weeks pregnant at time of survey | 26.1 (8.62; 4–41) |
| Edinburgh Postnatal Depression Survey (EPDS) score | 10.6 (5.33; 0–30) |
| | **Frequency (%)** |
| Race/ethnicity: | |
| White | 1,614 (87.0%) |
| Hispanic, Latino, or Spanish origin | 111 (5.98%) |
| Black or African American | 23 (1.24%) |
| Asian | 60 (3.23%) |
| American Indian or Alaskan Native | 12 (0.65%) |
| Other | 36 (1.94%) |
| Household income: | |
| < $49,999 | 221 (11.9%) |
| $50,000 - $99,999 | 610 (32.9%) |
| $100,000+ | 1,025 (55.2%) |
| Education level: | |
| Less than a bachelor's degree | 426 (23.0%) |
| Bachelor's degree | 644 (34.7%) |
| Degree beyond a bachelor's degree | 786 (42.4%) |
| Financial stress: | |
| Not worried | 1,062 (57.2%) |
| Worried | 794 (42.8%) |
| High-risk pregnancy: | |
| No | 1,209 (65.1%) |
| Yes | 647 (34.9%) |
| Exercise routine changed: | |
| No | 734 (39.6%) |
| Yes | 1,122 (60.4%) |
| RUCC code: | |
| Metro, 1 million+ | 1,128 (60.8%) |
| Metro, 250,000–999,999 | 394 (21.2%) |
| Metro, < 250,000 | 140 (7.54%) |
| Non-metro | 194 (10.5%) |

Sample means (with standard deviation and range) or frequency (percent) of model variables, for 1,856 participants included in the analyses.

Finally, women who reported their exercise routine had changed during the pandemic demonstrated significantly higher depression scores (B = 0.906, 95%CI: 0.423–1.39, p < 0.001). The first hypothesis was therefore accepted.

## Hypothesis 2

Logistic regression analyses were carried out to assess whether participant geographic location (i.e., non-metro vs. metro), as assessed by RUCC, was associated with significantly higher odds of reporting an altered exercise routine during the COVID-19 pandemic (Table 3). Participants who were farther along in their pregnancy had significantly lower odds of reporting changes to their exercise (OR = 0.985, 95%CI: 0.974–0.997, p = 0.010). Meanwhile, participants

**Table 2. Linear regression model assessing associations between reported exercise changes during the COVID-19 pandemic and Edinburgh Postnatal Depression Survey (EPDS) score.**

| Variable | B coefficient (SE, 95% CI) | p-value |
|---|---|---|
| Intercept | **12.4 (1.07, 10.4–14.5)** | **<0.001** |
| Age (years) | **-0.078 (0.036, -0.148-(-0.009))** | **0.028** |
| Weeks pregnant at time of survey | 0.024 (0.014, -0.003–0.051) | 0.085 |
| Race/ethnicity: | | |
| White | reference | |
| Hispanic, Latino, or Spanish origin | **1.32 (0.503, 0.332–2.30)** | **0.009** |
| Black or African American | 0.062 (1.07, -2.04–2.16) | 0.954 |
| Asian | -0.991 (0.672, -2.31–0.327) | 0.140 |
| American Indian or Alaskan Native | -0.340 (1.48, -3.24–2.56) | 0.818 |
| Other | -0.449 (0.858, -2.13–1.23) | 0.601 |
| Household income: | | |
| < $49,999 | reference | |
| $50,000 - $99,999 | **-0.861 (0.424, -1.69-(-0.029))** | **0.042** |
| $100,000+ | **-1.26 (0.446, -2.14-(-0.390))** | **0.005** |
| Education level: | | |
| Less than a bachelor's degree | reference | |
| Bachelor's degree | **-0.841 (0.349, -1.52-(-0.157))** | **0.016** |
| Degree beyond a bachelor's degree | **-1.08 (0.359, -1.79-(-0.379))** | **0.003** |
| Financial stress (no vs. yes) | **2.29 (0.246, 1.81–2.78)** | **<0.001** |
| High-risk pregnancy (no vs. yes) | **0.573 (0.290, 0.003–1.14)** | **0.049** |
| Exercise routine changed (no vs. yes) | **0.906 (0.246, 0.423–1.39)** | **<0.001** |

Beta coefficients are provided with standard errors, 95% confidence intervals, and p-values for each variable included in the model.

with a higher household income (compared to < $49,999) of $50,000 - $99,999 (OR = 1.58, 95%CI: 1.13–2.22, p = 0.008) or $100,000+ (OR = 1.62, 95%CI: 1.13–2.32, p = 0.008) and were more highly educated (compared to less than a bachelor's degree) with either a bachelor's degree (OR = 1.56, 95%CI: 1.19–2.06, p = 0.002) or a degree beyond a bachelor's degree (OR = 2.05, 95%CI: 1.54–2.72, p < 0.001) exhibited significantly higher odds of reporting exercise routine changes. Participants who reported experiencing pandemic-related financial stress (OR = 1.26, 95%CI: 1.03–1.54, p = 0.024) also displayed significantly higher odds of reporting exercise routine changes. Finally, compared to participants living in a non-metro area, those living in a metro area of 1 million or more people (OR = 1.99, 95%CI: 1.44–2.75, p < 0.001), a metro area of 250,000–999,999 people (OR = 1.75, 95%CI: 1.22–2.49, p = 0.002), or a metro area of < 250,000 people (OR = 2.06, 95%CI: 1.31–3.25, p = 0.002) demonstrated significantly higher odds of reporting exercise routine changes (Fig 2). The second hypothesis was consequently accepted.

## Discussion

The study findings support both hypotheses. Women who reported exercise changes during the COVID-19 pandemic exhibited significantly higher depression scores compared to those who reported no changes. Moreover, individuals living in metro areas of all sizes were significantly more likely to report exercise changes compared to women living in non-metro areas, potentially because women living in more rural areas felt safer venturing outside for walks or other forms of physical activity given the lower population density and less perceived risk of

**Table 3. Logistic regression model assessing associations between participant Rural-Urban Continuum Code (RUCC) and reported exercise change.**

| Variable | OR (SE, 95% CI) | p-value |
|---|---|---|
| Intercept | 0.441 (0.201, 0.180–1.08) | 0.073 |
| Age (years) | 1.00 (0.015, 0.972–1.03) | 0.981 |
| Weeks pregnant at time of survey | **0.985 (0.006, 0.974–0.997)** | **0.010** |
| Race/ethnicity: | | |
| White | reference | |
| Hispanic, Latino, or Spanish origin | 1.44 (0.309, 0.943–2.19) | 0.092 |
| Black or African American | 1.00 (0.451, 0.413–2.42) | 1.00 |
| Asian | 1.52 (0.467, 0.832–2.78) | 0.173 |
| American Indian or Alaskan Native | 1.28 (0.778, 0.390–4.21) | 0.682 |
| Other | 0.938 (0.331, 0.470–1.87) | 0.856 |
| Household income: | | |
| < $49,999 | reference | |
| $50,000 - $99,999 | **1.58 (0.272, 1.13–2.22)** | **0.008** |
| $100,000+ | **1.62 (0.296, 1.13–2.32)** | **0.008** |
| Education level: | | |
| Less than a bachelor's degree | reference | |
| Bachelor's degree | **1.56 (0.219, 1.19–2.06)** | **0.002** |
| Degree beyond a bachelor's degree | **2.05 (0.298, 1.54–2.72)** | **<0.001** |
| Financial stress (no vs. yes) | **1.26 (0.129, 1.03–1.54)** | **0.024** |
| High-risk pregnancy (no vs. yes) | 1.10 (0.132, 0.867–1.39) | 0.439 |
| RUCC: | | |
| Non-metro | reference | |
| Metro, 1 million+ | **1.99 (0.329, 1.44–2.75)** | **<0.001** |
| Metro, 250,000–999,999 | **1.75 (0.318, 1.22–2.49)** | **0.002** |
| Metro, < 250,000 | **2.06 (0.479, 1.31–3.25)** | **0.002** |

Odds ratios are provided with standard errors, 95% confidence intervals, and p-values for each variable included in the model.

viral exposure. Although cross-sectional and preliminary in nature, these results align with existing studies demonstrating the beneficial effects of exercise during pregnancy [13–20], suggesting moderate exercise may serve as a tool for supporting mental health during pregnancy.

Further, the association documented here suggests one possible mechanism by which the ongoing COVID-19 pandemic may impair maternal mental health. Business closures and stay at home orders resulting from the pandemic may directly inhibit one's ability to engage in physical activity. The majority of study participants (92.3%) indicated they were currently experiencing shelter in place orders, likely leading to disruptions in daily routines, including ability to engage in normal exercise activities. This idea is generally supported by preliminary analyses from the sample suggesting that a high percentage of participants (47.2%) explicitly reported exercising less following the onset of the pandemic, while many fewer respondents reported they were exercising more (9.15%). This difference suggests that exercise changes during the pandemic have generally resulted in less physical activity. This may be linked with gym, recreation center, and park closures that disrupt normal exercise routines, especially in urban areas where there is limited green space available for outdoor physical activity.

The disruption of exercise routines may have several negative effects contributing to elevated depression symptomatology, including decreased body satisfaction, increased physical

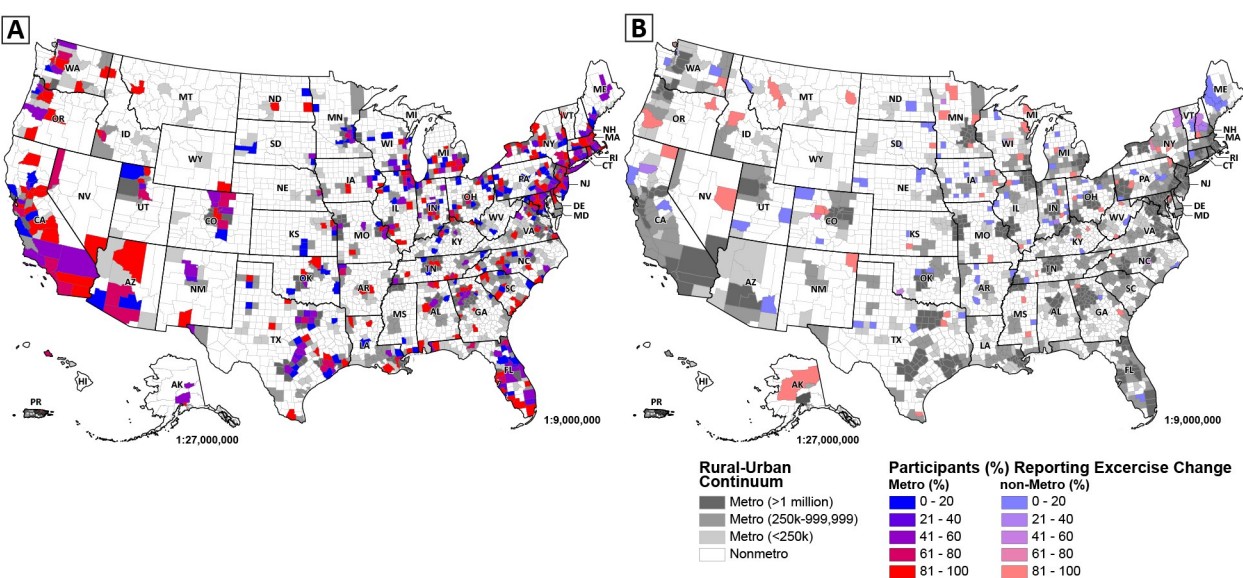

**Fig 2.** Percentage of study participants reporting change in exercise routine by United States county for (A) metro counties, and (B) non-metro counties. Map colors represent the proportion of participants in the sample reporting exercise routine changes during the pandemic within a given county (see legend), ranging from blue (a low percentage of sample participants reported exercise changes) to red (a high percentage of sample participants reported exercise routine changes). Metro counties are displayed in gray-scale according to the USDA Rural-Urban Continuum Codes (RUCC), while non-metro counties are displayed in white [35]. Reprinted from [35, 49] under a CC BY license, with permission from the U.S. Census Bureau, original copyright 2017, and the U.S. Department of Agriculture, original copyright 2013.

discomfort, perceptions of not being in control (of both physical changes associated with pregnancy and everyday activities), as well as poor mood [21–25]. Thus, maternal depression risk —already a concern among pregnant women prior to the COVID-19 pandemic—may be exacerbated during the pandemic partly because of disruptions to daily routines, including typical exercise regimens. The effects of altered exercise routines on maternal health may be especially relevant in the CARE study sample. These data were collected from women living in the U.S., the country hardest hit by the COVID-19 pandemic thus far [1]. The prolonged nature of the lockdowns and current uptick of infections in many parts of the country will likely continue to disrupt normal routines, including ability to exercise, for the foreseeable future.

Consequently, the association between depression score and changes in exercise routine documented here suggest that maternal depression screenings should account for the impact of COVID-19 on physical activity levels, in addition to other effects of the pandemic on maternal wellbeing. Additionally, providers should consider how ability to engage in physical activity during the pandemic may influence maternal mental health, particularly among women living in urban areas. Providers and/or fitness specialists could work with pregnant women and new mothers to find simple routines (e.g., exercises that can be completed safely from home without specialized equipment and which do not require much open space) to encourage physical activity during the pandemic across all demographics and living conditions. Indeed, it has been suggested that household chores, walks, gardening, and online fitness classes may be possible alternatives to support maternal physical and mental health [8].

However, efforts must be made to recommend exercises tailored to each individual's living conditions and physical health, as the negative effects of changed exercise routines are not likely to be experienced equally across all sociodemographic groups. Specifically, higher-income women with access to more resources may find it easier to adopt alternative forms of exercise as the pandemic continues (e.g., purchase home gym equipment or exercise videos),

allowing them stay physically active even if they live in urban areas where they cannot easily access a gym or exercise outside. Conversely, lower-income communities located in densely populated areas without infrastructure (e.g., green space and sidewalks) to support safe outdoor activities during COVID-19 may be more substantially impacted by pandemic-related public facility closures and shelter in place orders. However, this remains to be directly tested using a more diverse sample.

## Limitations

It should be noted that despite the strengths of these analyses (e.g., large sample size and participants from across the country), several important study limitations exist. First, this study is cross-sectional, making it difficult to establish causality. For instance, it is not possible to definitively establish whether the significant relationship observed between exercise routine change during the pandemic and depression score is due to disrupted exercise routines leading to later depression or to maternal depression leading to decreased motivation to exercise. The relationship between physical activity and depression appears to be bidirectional in nature, with exercise protecting against depression but baseline depression also subsequently decreasing physical activity levels [50]. Longitudinal data collection is needed to establish these causal relationships.

In addition, due to the use of convenience sampling, these data are not representative of the U.S. population as a whole; white, educated, wealthy women are overrepresented in the present sample [51]. Additional work is needed to determine whether the associations observed here are also evident across a more representative, diverse sample of the U.S. population. Finally, quantitative data on pre-pandemic exercise patterns were not collected, making it difficult to establish whether the patterns documented here partly reflect pre-existing physical activity differences. For example, previous work indicates that pregnant women living in urban areas exhibit higher levels of moderate exercise compared to those living in rural areas [29, 30]; women living in non-metro areas may therefore be less likely to report exercise changes as they were not regularly exercising prior to the pandemic. Still, this pattern does not detract from the finding that women living in metro areas appear to be exercising less during the pandemic, which may negatively impact their physical and mental wellbeing.

## Conclusions

The COVID-19 pandemic has far-ranging effects on both mental and physical health. Certain segments of the population may be especially impacted by the pandemic; including pregnant women, who already suffer from an elevated depression risk compared to the general public [9]. The findings presented here support previous work and suggest an association exists between ability to maintain an exercise routine during pregnancy and depression risk. Providers should explicitly ask patients how the pandemic has impacted their exercise routines and consider this a risk factor for depression. Moreover, efforts should be made to recommend exercises that are tailored to individual living conditions (e.g., whether they can safely exercise outdoors) and physical health.

Specifically, the findings presented here indicate that women living in metro areas were significantly more likely to report exercise routine changes during the pandemic than women living in non-metro areas. This may signify an opportunity for interventions—especially in urban areas—which rely on simple exercises that can safely be performed at home without specialized equipment (e.g., squats, lunges, side-lying leg lifts, etc.). Importantly, exercise recommendations may be most effective if communicated by providers who understand the personal health and space limitations of each individual patient. Exercise represents a potential non-

pharmaceutical tool to support both mental and physical health among pregnant women, both during and after the pandemic.

## Supporting information

**S1 Checklist. STROBE checklist.**
(DOCX)

## Acknowledgments

We would like to thank Nadia Clement for her assistance building the online survey and designing the recruitment flyers. We would also like to thank Margaret Sherin, Gloriuese Uwizeye, and Chlöe Sweetman for their valuable feedback on the questionnaire during development. Further, we express our gratitude to Grace Alston, Cecily Craighead, Amanda Lu, and Rebecca Milner for their assistance in data coding. Finally, we thank Matthew Nagy for his assistance in data cleaning.

## Author Contributions

**Conceptualization:** Theresa E. Gildner.

**Data curation:** Theresa E. Gildner.

**Formal analysis:** Theresa E. Gildner, Elise J. Laugier.

**Funding acquisition:** Theresa E. Gildner, Zaneta M. Thayer.

**Investigation:** Theresa E. Gildner, Zaneta M. Thayer.

**Methodology:** Theresa E. Gildner, Zaneta M. Thayer.

**Project administration:** Theresa E. Gildner, Zaneta M. Thayer.

**Visualization:** Elise J. Laugier.

**Writing – original draft:** Theresa E. Gildner.

**Writing – review & editing:** Theresa E. Gildner, Elise J. Laugier, Zaneta M. Thayer.

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
