## [Decision Letter · Decision Letter 0]

9 Nov 2020

PONE-D-20-29604

Exercise routine change is associated with prenatal depression scores during the COVID-19 pandemic among pregnant women across the United States

PLOS ONE

Dear Dr. Gildner,

Thank you for submitting your manuscript to PLOS ONE. After careful consideration, we feel that it has merit but does not fully meet PLOS ONE’s publication criteria as it currently stands. Therefore, we invite you to submit a revised version of the manuscript that addresses the points raised during the review process.

We look forward to receiving your revised manuscript.

Kind regards,

Vijayaprasad Gopichandran

Academic Editor

PLOS ONE

Journal Requirements:

2.We note that you have indicated that data from this study are available upon request. PLOS only allows data to be available upon request if there are legal or ethical restrictions on sharing data publicly. For more information on unacceptable data access restrictions, please see http://journals.plos.org/plosone/s/data-availability#loc-unacceptable-data-access-restrictions.

3.We note that [Figure(s) 1 and 2] in your submission contain [map/satellite] images which may be copyrighted. All PLOS content is published under the Creative Commons Attribution License (CC BY 4.0), which means that the manuscript, images, and Supporting Information files will be freely available online, and any third party is permitted to access, download, copy, distribute, and use these materials in any way, even commercially, with proper attribution. For these reasons, we cannot publish previously copyrighted maps or satellite images created using proprietary data, such as Google software (Google Maps, Street View, and Earth). For more information, see our copyright guidelines: http://journals.plos.org/plosone/s/licenses-and-copyright.

1.    You may seek permission from the original copyright holder of Figure(s) [1 and 2] to publish the content specifically under the CC BY 4.0 license. 

Reviewers' comments:

Reviewer's Responses to Questions

**Comments to the Author**

1. Is the manuscript technically sound, and do the data support the conclusions?

Reviewer #1: Yes

Reviewer #2: Yes

Reviewer #3: Partly

Reviewer #4: Partly

2. Has the statistical analysis been performed appropriately and rigorously? 

Reviewer #1: Yes

Reviewer #2: Yes

Reviewer #3: Yes

Reviewer #4: I Don't Know

3. Have the authors made all data underlying the findings in their manuscript fully available?

Reviewer #1: No

Reviewer #2: Yes

Reviewer #3: No

Reviewer #4: No

4. Is the manuscript presented in an intelligible fashion and written in standard English?

Reviewer #1: Yes

Reviewer #2: Yes

Reviewer #3: Yes

Reviewer #4: Yes

5. Review Comments to the Author

Reviewer #1: The current manuscript aims at investigating whether the change in an exercise routine during the Covid-19 pandemic is associated with the prenatal depression scores in pregnant women in the United States. Moreover, the current study also aims to investigate whether women in metro areas (urban) are more likely to report changes in exercise routines as compared to those in the non-metro areas. The authors collected data from an online survey of pregnant women in the United States. A total of 1,862 responses were collected. The findings revealed that women reporting exercise changes during the pandemic also reported significantly higher depression scores compared to those reporting no changes. Additionally, women living in metro areas were significantly more likely to report exercise changes compared to women living in non-metro areas. The authors suggested that an association between the change in an exercise routine and depression scores exist. They concluded that moderate exercise may represent a non-pharmaceutical tool for supporting maternal physical and mental health.

An adequate background about the association between the physical activity and mental health of pregnant women is provided, although the question of pandemic related exercise changes and mental health in pregnant women is relatively new. The manuscript is well written, and the study is interesting. Overall, the writing is coherent and concise.

The data set is quite large and collected from all across the United States. The methods and data analysis are clear and thorough. The methods are well-written and extensively explained. The results and data interpretation are clear and interesting. The objectives are met and the conclusions are supported by the data provided.

Discussion is clear and leaves the reader with a good understanding of how changes in an exercise routine have an influence on mental health of pregnant women and also provides future considerations. Finally, the authors suggested that the exercise routine (moderate aerobic exercise) can serve as a non-pharmaceutical treatment in order to avoid the harmful exposure of offspring to psychotropic medications taken by mother during pregnancy.

Overall, the manuscript is well written and strong, and adds to the existing literature of Covid-19 and mental health.

Reviewer #2: The abstract is clear and well explained. All sections are well presented.

The introduction section provides background information on COVID-19 situation and its impact in the world and describes how COVID-19 has posed a risk to pregnancy. It also explains the mental status of pregnant women during the COVID-19 pandemic. Necessary evidences have been discussed. It also discusses the elevation of maternal depression during the epidemic and the use of alternative methods for its treatment which includes regular exercise. Overall the introduction section is well described.

The material and methods section is divided into different subsections. The design has been explained well along with description of all the variables used. It also explains the use of data from the CARE study. The statistical analysis are also fine.

The results are described sufficiently. The characteristics of the participants are well described. Tables have been presented to show the descriptive characteristics. Linear regression analysis has been used to test the hypothesis. Results are fine.

The discussions are sufficient and supports the findings made from the study.

The limitations of the study is also well presented.

The conclusions are in line with the findings and the discussions made.

References are valid and sufficient.

Reviewer #3: The article contains a lot of useful information on the issue. The topic is very interesting and use of sources is appropriate. Some revisions are necessary: The aim of the study is clear, the title is informative and relevant, but the abstract section involve too much information. The authors should not write such details about their manuscript. As regards the references, are relevant, but not so recent, referenced correctly, and there are appropriate key studies included. However, the authors should add some more, in order to establish their findings.

Introduction/ background

It is quite clear what is already known about this topic and the research question is clearly outlined. The research question is not justified clearly, given what is already known about the topic. Some chapters are lengthy related to others. The authors have to keep uniformity about the length of the chapters.

Overall

The article contains a lot of useful information and the topic is very interesting. Some revisions are necessary: The author should provide more information in some chapters and omit some others. In addition, they should conclude a chapter with the limitations of the studies mentioned.

Reviewer #4: GENERAL:

The authors have presented data, which is very poignant at this time, therefore I encourage the authors to revise and resubmit.

1. Is the manuscript technically sound, and do the data support the conclusions? Until the clearly state why they have chosen logistic regression, it cannot be determined. Also, as is pointed out in the attached PDF, the hypotheses are formally stated; hence, the results/conclusions do not flow from the hypotheses and analyses.

2. Has the statistical analysis been performed appropriately and rigorously? The authors do test the underlying conditions for the logistic regression, however, as mentioned above in #1., until the authors justify the test, this cannot be determined.

3. Have the authors made all data underlying the findings in their manuscript fully available? The authors have explained.

4. There are a few grammatical errors. See attached PDF. In general, proofing is needed.

ALSO:

SPECIFIC RECOMMENDATIONS

Title:

1.Exercise routine change is associated with prenatal depression scores during the COVID-19 pandemic among pregnant women across the United States – suggested: “Exercise routine change is associated with prenatal depression during the COVID-19 pandemic among pregnant women in the United States”

Abstract:

1.Methods: Kindly place all in past-tense “used” instead of “uses”; insert “reported” before “COVID-19-related”

2.Conclusions do not follow from hypotheses.

Background:

-Sentence requires a citation—line 97, “given the effects of shelter in place orders”, can be qualified with U.S. data as of the writing of the manuscript. i.e. the number of states with or without shelter in place / stay at home orders, etc.

-Please include a justification for all of your covariates in the background, e.g. age,

Methods:

What is stated, is a more general wording of the hypotheses, which should be included earlier on – in the Abstract as well as at the end of the Introduction:

“Hypothesis 1: Linear regression analyses assessed whether exercise routine change during the

170 COVID-19 pandemic was significantly associated with participant depression score.

171 Hypothesis 2: Logistic regression analyses tested whether a participant RUCC classification of

172 “metro” was linked with significantly higher odds of reporting an altered exercise routine during the

173 pandemic compared to a classification of “non-metro”.

Please review how hypotheses should be stated, and re-state the hypotheses more formally here in the Methods section.

Statistical Analysis

Inform the audience by explaining why LR is the most appropriate test, with citations from a methods book or published article.

Results:

Hypotheses must be accepted or rejected.

6. PLOS authors have the option to publish the peer review history of their article (what does this mean?). If published, this will include your full peer review and any attached files.

Reviewer #1: No

Reviewer #2: No

Reviewer #3: **Yes: **Dr. Kalliopi Megari

Reviewer #4: **Yes: **Lunthita M. Duthely, Ed.D.

---

## [Author Response · Author response to Decision Letter 0]

16 Nov 2020

Response to Editor

Thank you for drawing our attention to the following journal requirements. We have addressed these issues and respond to each specific point below.

*Response: We apologize for any formatting errors. We have gone through the document and updated the formatting to comply with PLOS ONE style requirements, as outlined in the shared links.

2.We note that you have indicated that data from this study are available upon request. PLOS only allows data to be available upon request if there are legal or ethical restrictions on sharing data publicly. For more information on unacceptable data access restrictions, please see http://journals.plos.org/plosone/s/data-availability#loc-unacceptable-data-access-restrictions.

*Response: We have updated the cover letter with this information, as requested.

3.We note that [Figure(s) 1 and 2] in your submission contain [map/satellite] images which may be copyrighted. All PLOS content is published under the Creative Commons Attribution License (CC BY 4.0), which means that the manuscript, images, and Supporting Information files will be freely available online, and any third party is permitted to access, download, copy, distribute, and use these materials in any way, even commercially, with proper attribution. For these reasons, we cannot publish previously copyrighted maps or satellite images created using proprietary data, such as Google software (Google Maps, Street View, and Earth). For more information, see our copyright guidelines: http://journals.plos.org/plosone/s/licenses-and-copyright.

1. You may seek permission from the original copyright holder of Figure(s) [1 and 2] to publish the content specifically under the CC BY 4.0 license. 

*Response: Thank you for raising this point. We have edited the figure captions to clarify that the maps were created using open domain data. Specifically, the last sentence of each figure caption now reads: “Reprinted from (35,49) under a CC BY license, with permission from the U.S. Census Bureau, original copyright 2017, and the U.S. Department of Agriculture, original copyright 2013.”

35. United States Department of Agriculture (USDA). Rural-Urban Continuum Codes [Internet]. 2013 [cited 2020 Jul 2]. Available from: https://www.ers.usda.gov/data-products/rural-urban-continuum-codes.aspx

49. OpenStreetMap. TIGER/Line Shapefile, 2017, nation, U.S., Current County and Equivalent National Shapefile. [Internet]. 2017 [cited 2020 Nov 12]. Available from: https://catalog.data.gov/dataset/tiger-line-shapefile-2017-nation-u-s-current-county-and-equivalent-national-shapefile

Response to Reviewers

We thank the reviewers for their thoughtful comments. We have revised the manuscript based on their feedback and believe that we have produced an improved submission. All changes in the manuscript have been marked using track changes and we address the individual reviewer comments below.

Reviewer #1: The current manuscript aims at investigating whether the change in an exercise routine during the Covid-19 pandemic is associated with the prenatal depression scores in pregnant women in the United States. Moreover, the current study also aims to investigate whether women in metro areas (urban) are more likely to report changes in exercise routines as compared to those in the non-metro areas. The authors collected data from an online survey of pregnant women in the United States. A total of 1,862 responses were collected. The findings revealed that women reporting exercise changes during the pandemic also reported significantly higher depression scores compared to those reporting no changes. Additionally, women living in metro areas were significantly more likely to report exercise changes compared to women living in non-metro areas. The authors suggested that an association between the change in an exercise routine and depression scores exist. They concluded that moderate exercise may represent a non-pharmaceutical tool for supporting maternal physical and mental health.

An adequate background about the association between the physical activity and mental health of pregnant women is provided, although the question of pandemic related exercise changes and mental health in pregnant women is relatively new. The manuscript is well written, and the study is interesting. Overall, the writing is coherent and concise.

The data set is quite large and collected from all across the United States. The methods and data analysis are clear and thorough. The methods are well-written and extensively explained. The results and data interpretation are clear and interesting. The objectives are met and the conclusions are supported by the data provided.

Discussion is clear and leaves the reader with a good understanding of how changes in an exercise routine have an influence on mental health of pregnant women and also provides future considerations. Finally, the authors suggested that the exercise routine (moderate aerobic exercise) can serve as a non-pharmaceutical treatment in order to avoid the harmful exposure of offspring to psychotropic medications taken by mother during pregnancy.

Overall, the manuscript is well written and strong, and adds to the existing literature of Covid-19 and mental health.

*Response: Thank you very much for the positive feedback.

Reviewer #2: The abstract is clear and well explained. All sections are well presented.

The introduction section provides background information on COVID-19 situation and its impact in the world and describes how COVID-19 has posed a risk to pregnancy. It also explains the mental status of pregnant women during the COVID-19 pandemic. Necessary evidences have been discussed. It also discusses the elevation of maternal depression during the epidemic and the use of alternative methods for its treatment which includes regular exercise. Overall the introduction section is well described.

The material and methods section is divided into different subsections. The design has been explained well along with description of all the variables used. It also explains the use of data from the CARE study. The statistical analysis are also fine.

The results are described sufficiently. The characteristics of the participants are well described. Tables have been presented to show the descriptive characteristics. Linear regression analysis has been used to test the hypothesis. Results are fine.

The discussions are sufficient and supports the findings made from the study.

The limitations of the study is also well presented.

The conclusions are in line with the findings and the discussions made.

References are valid and sufficient.

*Response: We very much appreciate your feedback, thank you.

Reviewer #3: The article contains a lot of useful information on the issue. The topic is very interesting and use of sources is appropriate. Some revisions are necessary: The aim of the study is clear, the title is informative and relevant, but the abstract section involve too much information. The authors should not write such details about their manuscript. 

*Response: We have trimmed some of the explanations provided in the introduction of the abstract.

As regards the references, are relevant, but not so recent, referenced correctly, and there are appropriate key studies included. However, the authors should add some more, in order to establish their findings.

*Response: As requested, we have added additional recent citations to better establish the study and our results.

Introduction/ background: It is quite clear what is already known about this topic and the research question is clearly outlined. The research question is not justified clearly, given what is already known about the topic. Some chapters are lengthy related to others. The authors have to keep uniformity about the length of the chapters.

*Response: Thank you for this suggestion, we have added some more details to better justify the research question. We have also moved around a couple sentences to make the paragraphs more uniform in length. 

Overall: The article contains a lot of useful information and the topic is very interesting. Some revisions are necessary: The author should provide more information in some chapters and omit some others. In addition, they should conclude a chapter with the limitations of the studies mentioned.

*Response: We have added some recent citations, as noted above. We have also more explicitly highlight the limitations of previous work in the sentence justifying this study (i.e., that previous studies have not directly tested the links between pandemic-related changes in exercise routines and depression risk, or whether geographic location may influence the likelihood of exercise disruptions in the first place).

Reviewer #4: 

The authors have presented data, which is very poignant at this time, therefore I encourage the authors to revise and resubmit.

1. Is the manuscript technically sound, and do the data support the conclusions? Until the clearly state why they have chosen logistic regression, it cannot be determined. Also, as is pointed out in the attached PDF, the hypotheses are formally stated; hence, the results/conclusions do not flow from the hypotheses and analyses.

*Response: We have added a statement explicitly explaining why regression analyses were used in the Methods section. We have also rephrased the hypotheses and conclusions as suggested (to ensure that the results and conclusions flow from the hypotheses and analyses).

2. Has the statistical analysis been performed appropriately and rigorously? The authors do test the underlying conditions for the logistic regression, however, as mentioned above in #1., until the authors justify the test, this cannot be determined.

*Response: We have added the justification for the use of regression analyses at the end of the Methods section. 

3. Have the authors made all data underlying the findings in their manuscript fully available? The authors have explained.

*Response: We have added this information to the cover letter, explaining how the sensitive nature of these data prevent us from uploading them to a public repository (e.g., depression scores and zip code data that may be potentially identifiable for participants living in low population density areas, when combined with other covariate data). Instead, data will be shared with qualified researchers through a request form available on the CARE Study website. 

4. There are a few grammatical errors. See attached PDF. In general, proofing is needed.

*Response: We have made all of the edits suggested in the PDF and read through the manuscript looking for typos. Thank you for these suggestions.

ALSO: SPECIFIC RECOMMENDATIONS

Title: 1.Exercise routine change is associated with prenatal depression scores during the COVID-19 pandemic among pregnant women across the United States – suggested: “Exercise routine change is associated with prenatal depression during the COVID-19 pandemic among pregnant women in the United States”

*Response: We have retained the word “scores” in the title because we do not want to imply that these women have been diagnosed with depression. We are analyzing depression scores (measured using a gold standard scale), not medical depression diagnoses.

Abstract: 1.Methods: Kindly place all in past-tense “used” instead of “uses”; insert “reported” before “COVID-19-related”

*Response: Thank you for catching this, we have made these edits. 

2.Conclusions do not follow from hypotheses.

*Response: Thank you for raising this point. We have rephrased the conclusions so that they better follow the hypotheses.

Background:

-Sentence requires a citation—line 97, “given the effects of shelter in place orders”, can be qualified with U.S. data as of the writing of the manuscript. i.e. the number of states with or without shelter in place / stay at home orders, etc.

*Response: We have clarified that shelter in place orders were evident nationwide to at least some extent (e.g., on continuum ranging from all individuals ordered to shelter in place, to only certain groups of individuals or counties being placed under these orders) throughout the spring (when the participants completed the survey) and added a citation for those interested in further exploring the timeline of shelter in place restrictions by state. 

-Please include a justification for all of your covariates in the background, e.g. age,

*Response: We have added these justifications in the Methods section. Specifically, when we describe how each variable was measured, we have added a justification for why that variable was included during statistical analyses. 

Methods: What is stated, is a more general wording of the hypotheses, which should be included earlier on – in the Abstract as well as at the end of the Introduction:

“Hypothesis 1: Linear regression analyses assessed whether exercise routine change during the

170 COVID-19 pandemic was significantly associated with participant depression score.

171 Hypothesis 2: Logistic regression analyses tested whether a participant RUCC classification of

172 “metro” was linked with significantly higher odds of reporting an altered exercise routine during the

173 pandemic compared to a classification of “non-metro”.

Please review how hypotheses should be stated, and re-state the hypotheses more formally here in the Methods section.

*Response: As requested, we have switching the wording of the hypotheses. Moving the more formal wording to the Methods section and the less formal wording to the Abstract and Introduction. 

Statistical Analysis: Inform the audience by explaining why LR is the most appropriate test, with citations from a methods book or published article.

*Response: We have added this information and citations at the end of the Methods section.

Results: Hypotheses must be accepted or rejected.

*Response: We have altered the wording in the Results section to make it clearer that both hypotheses were accepted (i.e., cut “as hypothesized” to instead say “the hypothesis was therefore accepted”).

---

## [Editor Report · Decision Letter 1]

18 Nov 2020

Exercise routine change is associated with prenatal depression scores during the COVID-19 pandemic among pregnant women across the United States

PONE-D-20-29604R1

Dear Dr. Gildner,

We’re pleased to inform you that your manuscript has been judged scientifically suitable for publication and will be formally accepted for publication once it meets all outstanding technical requirements.

Kind regards,

Vijayaprasad Gopichandran

Academic Editor

PLOS ONE
---

## [Editor Report · Acceptance letter]

20 Nov 2020

PONE-D-20-29604R1 

Exercise routine change is associated with prenatal depression scores during the COVID-19 pandemic among pregnant women across the United States 

Dear Dr. Gildner:

I'm pleased to inform you that your manuscript has been deemed suitable for publication in PLOS ONE. Congratulations! Your manuscript is now with our production department. 

Kind regards, 

on behalf of

Dr. Vijayaprasad Gopichandran 

Academic Editor

PLOS ONE